# The Generation of Gridded Emissions Data for CMIP6

Leyang Feng[1], Steven J Smith[1], Caleb Braun[1], Monica Crippa[2], Matthew J. Gidden[3], Rachel Hoesly[1], Zbigniew Klimont[3], Margreet van Marle[4], Maarten van den Berg[5], Guido R. van der Werf[6]

[1] Joint Global Change Research Institute, Pacific Northwest National Laboratory, College Park, MD, USA
[2] European Commission, Joint Research Centre (JRC), Ispra, Italy
[3] International Institute for Applied Systems Analysis, Laxenburg, Austria
[4] Deltares, Delft, the Netherlands
[5] PBL Netherlands Environmental Assessment Agency, The Hague, The Netherlands
[6] Faculty of Science, Vrije Universiteit Amsterdam, Amsterdam, the Netherlands

*Correspondence to*: Steven Smith (ssmith@pnnl.gov)

## Abstract

Spatially distributed anthropogenic and open burning emissions are fundamental data needed by Earth system models. We describe the methods used for generating gridded data sets produced for use by the modelling community, particularly for the Coupled Model Inter-comparison Project Phase 6. The development of three sets of gridded data for historical open burning, historical anthropogenic, and future scenarios were coordinated to produce consistent data over 1750-2100. Historical data up to 2014 were provided with annual resolution and future scenario data in 10-year intervals. Emissions are provided on a sectoral basis, along with additional files for speciated non-Methane Volatile Organic Compounds (NMVOCs). An automated framework was developed to produce these datasets to ensure that they are reproducible and facilitate future improvements. We discuss the methodologies used to produce these data along with limitations and potential for future work.

## 1 Introduction

Anthropogenic activities, from the generation of electricity to the ignition of forest fires, result in the emissions of gases and aerosol species into the atmosphere. These emissions in turn, alter atmospheric composition, deposition rates, and the Earth's radiative balance. Emissions have, for the most part, increased over the 20th century, leading to higher aerosol concentrations and tropospheric ozone levels, as industrial activities and fossil fuel consumption increased. Over the past couple decades air pollutant emissions have shifted away from North America and Europe, due in large part to air pollution controls, to East and South Asia, driven by rapid industrialization and population growth in those regions.

One of the important tools used to examine these impacts are models of the Earth system, including chemical-transport, climate-chemistry, and Earth-system models. These models require girded emissions data in order to simulate the impact of these emissions within the Earth system. Furthermore. for  multi-decadal to century long model runs, temporally and spatially consistent emissions data are needed. Here we describe the production of several related gridded datasets that were produced, in large part, to facilitate the Coupled Model Inter-comparison Project Phase 6 (CMIP6; Eyring et al. 2016). These datasets contain anthropogenic chemically reactive gases (CO, $CH_4$, $NH_3$, $NO_X$, $SO_2$, NMVOC), carbonaceous aerosols (BC and OC) – the principal climatically important primary particle emission species, and fossil $CO_2$ emissions, with non-methane organic hydrocarbons (NMVOCs) also provided as separate species (Table 1). These emission data products are needed to drive emissions-driven simulations over both historical and future time periods for ScenarioMIP, AerChemMIP, and C4MIP (Eyring et al. 2016, Collins et al. 2017, Jones et al. 2016). Using these emission datasets, models can estimate atmospheric concentrations of primary and secondary fine particulates and tropospheric ozone, which impact human health, ecosystem functioning, and the Earth's radiative balance.

The first dataset considered here are anthropogenic emissions, which are defined as emissions that stem directly from human activities such as energy transformation, buildings, transportation, and agricultural and industrial activities

(see Hoesly et al. 2018 for a complete listing). Historical emissions gridding is also discussed in Hoesly et al. (2018), along with an extensive description of the methodologies used to produce country-level historical anthropogenic emissions. We provide here a more complete discussion of the gridding methodology, a description of the closely related methods used for gridding future emissions, and related supplementary data such as speciated VOCs.

Compared to previous datasets, our anthropogenic data has a greater degree of consistency across species and over time, with seasonality for all species. The historical anthropogenic emissions were produced by the Community Emissions Data System (CEDS). This paper focuses on the data produced for CMIP6 (see Appendix section A.1). Updated emission data releases for general scientific use by the CEDS project are in progress.

The second dataset is historical open burning emissions which are defined as forest, grassland, and peatland fires,
along with agricultural waste burning (AWB) on fields. While open burning emissions can also have anthropogenic drivers, these emissions are in a separate category here, and elsewhere in the literature, as the techniques for estimating are generally distinct from the methods used for "anthropogenic" emissions. The open burning emissions over recent years are from the Global Fire Emissions Database version 4 with small fires (GFED4s, Van der Werf et al 2017), which are driven by satellite data since 1997. Estimates for earlier years are based on proxies and fire
models. We only briefly discuss this dataset in this paper since those data are inherently in gridded form during their development, as described in detail by van Marle et al. (2017). Note that open burning emissions are often described in the literature as "biomass burning", but we do not use this term to avoid confusion with anthropogenic biofuel combustion such as biofuel use in cookstoves, which is included in the anthropogenic emissions dataset.

The third set of data are gridded data over the future (2015-2100) for these same species for both anthropogenic and
open burning sectors for selected future scenarios. These future gridded data were produced using a variation of the same gridding methodologies used for the historical anthropogenic data, used to produced gridded data for both anthropogenic and open burning emissions, which is why these are discussed together in this paper. The future emission trajectories are discussed in Gidden et al. (2018), with gridding methodology described in more detail herein. As discussed below, the future gridded emissions builds on these two historical datasets and, in large part,
inherits their properties such as within-country spatial distribution and seasonality.

A number of global gridded data sets have been produced over the years. One of the most widely used datasets is the Emissions Database for Global Atmospheric Research (EDGAR), which provides an independent estimate of historical greenhouse gas (GHG) and pollutant emissions by country, sector, and spatial grid ($0.1 \times 0.1$ degree, Crippa et al., 2016; EC-JRC/PBL, 2016). The most recent release provides annual and monthly data over 1970 –
2012 (Crippa et al. 2018). The GAINS (Greenhouse gas - Air pollution Interactions and Synergies) model (Amann et al., 2011) has been used to produce regional and global emission estimates for recent years (1990- 2010; in five year intervals). These estimates formed the basis for ECLIPSE emission data also available in gridded form (Klimont et al., 2017). Other global datasets include the HTAP v2 (Janssens-Maenhout et al., 2015) emissions data used by the Task Force on Hemispheric Transport of Air Pollutants (HTAP). That dataset merged EDGAR with regional and
country-level gridded emissions data for 2008 and 2010.

Lamarque et al. (2010) developed the historical data set used in the Coupled Model Intercomparison Project Phase 5 (CMIP5), which included global, gridded estimates of anthropogenic and open burning emissions from 1850 – 2000 at 10 year intervals. It was a compilation of "best available estimates" from many sources including EDGAR-HYDE (van Aardenne et al., 2001), RETRO (Schultz and Sebastian, 2007) and emissions reported by, largely, Organization
for Economic Co-operation and Development (OECD) countries over recent years. One focal point of that work was the compilation of a year 2000 emissions dataset that was used as the starting point for the future projections. See Hoesly et al. (2018) and van Marle et al. (2017) for a comparison of the CMIP6 and CMIP5 anthropogenic and open burning datasets respectively, and Gidden et al. (2019) for a comparison of the CMIP6 and CMIP5 projections.

We first discuss the overall methodologies for producing the gridded data, then present the gridded data and discuss
the underlying properties of this data, focusing on the anthropogenic sources. The paper concludes with a discussion of issues identified in these data and potential further work to improve their quality.

## 2 Data and Methodology

### 2.1 Methodology overview

We first provide an overview of the gridding methodology, with further details provided in the following sub-sections. The gridding methodology for historical anthropogenic and all future emissions is summarized in Figure 1.
The fundamental underlying data used here are emissions by country and sector. This provides both total emission trends over time as well as changes in the sectoral composition of each emission species. As discussed in Hoesly et al. (2018), emissions in the pre-industrial period were generally dominated by biofuel use in the residential sector. As industrialization proceeded emissions from industrial, energy transformation, and transportation sectors became increasingly important. Emissions are translated to a spatial grid for each country and aggregate gridding sector (see
Table 2). The methodology applied to future emissions is similar to that used for historical emissions, although with a lower sectoral and temporal resolution, as detailed below. This comprehensive gridded dataset, therefore, is produced with a consistent methodology by sector across all emission species spanning 1750 through 2014, with a consistent set of future projections over 2015 - 2100. Details of the methodology are first described for the historical emissions data (1750-2014), followed by a discussion of areas where the methodology differs for future emissions (2015-2100).
Further details including code are available on-line for both historical and future gridding (§ 5 below).

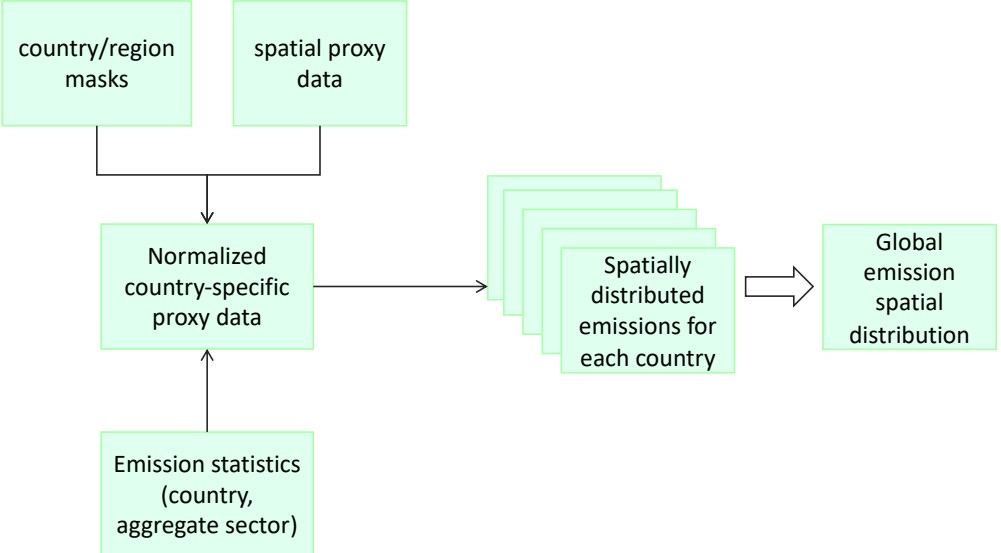

Figure 1: Emissions gridding overview. Emissions data at the level of country and aggregate sector are mapped to spatial grids separately by country and sector, and then combined into global emissions grids as described in the main text.

### 2.2 Historical Anthropogenic Emissions

Aggregate anthropogenic emissions over the historical period by country and CEDS sector from Hoesly et al. (2018) are aggregated to 16 intermediate sectors (Table 2) and mapped to a 0.5 × 0.5 degree grid as described below. The intermediate sectors were selected according to the availability of gridded proxy data in order to provide as much spatial detail as possible. For final distribution, gridded emissions are further aggregated to 9 sectors: agriculture,
energy, industrial, transportation, residential/commercial/other, solvents, waste, international shipping, and aircraft. These final sectors were chosen to be consistent with previous practice (e.g., Lamarque et al. 2010), which includes matching the generally lower level of sectoral disaggregation available from future projections, and to reduce overall file size. Gridded emissions data are produced as CF-compliant NetCDF files (http://cfconventions.org/) and converted to flux (kg $m^{-2}s^{-1}$). Data were aggregated into a number of multi-year files, keeping individual file sizes
under about 1 Gb. Filenames and netCDF metadata are generated so that files are ready for distribution using input4MIPS on the Earth System Grid Federation system.

Emissions at the level of country and gridding sector are mapped to a spatial grid using a variety of spatial proxy data, as described below. Emissions are distributed into target grid cells $XY$ as:

$$emission_{XY} = emission_c \times \frac{proxy\_value_{XY}}{\sum proxy\_value_{XY}}, \tag{1}$$

where $emission_{XY}$ is the emissions value assigned to coordinate $XY$; $emission_c$ is the total country-level emissions
for the gridding sector; $proxy\_value_{XY}$ is the value of the proxy data for the cell $XY$, and the sum is performed over all coordinates $XY$ that are within the specified country.

The proxy data values are pre-processed for each country by multiplying the data by the area fraction of each grid cell that is in the specified country at an annual resolution (if annual gridded data are available). In this way grid cells that contain multiple countries will be assigned emissions proportionately. After assigning emissions to spatial grids
by country, emissions from each country, are then spatially added into a global matrix. The "country" list includes a global region for emissions related to international shipping which are not associated with a particular country. Aircraft emissions are gridded separately, with one three-dimensional distribution that is scaled uniformly to match the global emissions estimate.

In most cases the proxy data are gridded emissions, so as not to duplicate the effort needed to convert raw proxy data
into the form needed for emissions inventories. We define two levels of spatial proxy data, the primary gridding proxies, and a backup gridding proxy, which is gridded population. The backup gridding proxy is used where the primary gridding proxies are not appropriate due to either: (1) the primary proxy being not available (e.g. is equal to zero) for the given country-sector-year combination or (2) the primary proxy is inaccurate for the given country-sector combination. In the latter case, we perform this proxy substitution when the ratio of proxy to sector emissions
data for that country-sector combination is an outlier compared to the global distribution of this ratio across countries.

Over recent decades the primary gridding proxy data were from the EDGAR v4.2 (EC-JRC/PBL, 2012) inventory (Table 2) since this data was available over 1970 through 2008. Road transportation uses the EDGAR 4.3.2 road transportation grid, which is significantly improved over previous versions (Crippa et al., 2016), but was only
available for 2010 at the time these data were produced, so the country-specific 2010 spatial distribution is used for all years. Flaring emissions use a blend of grids from EDGAR and ECLIPSE (Klimont et al., 2017). The backup gridded population proxy, as well as the proxy for early years for the residential/commercial sector, is a combination of gridded population from Gridded Population of the World (GPW) (Doxsey-Whitfield et al., 2015) and HYDE (Goldewijk et al., 2011). Aircraft emission distributions are from Lee et al. (2009) and international shipping uses
ECLIPSE shipping grids, with additional data from Endresen et al. (2003) for NMVOC emissions from oil tanker venting as used in Lamarque et al. (2010).

The only proxy data that vary before 1970 are those for the RCO (residential, commercial, other) and waste sectors (Table 2). For the RCO sector, for 1900 to 1969, the proxy linearly blends grids from EDGAR v4.2 RCO 1970 grid and gridded population, with gridded population used for years before 1900. The proxy for the waste burning sector
is based on rural population. More specific methodological details for certain aspects of the emissions dataset are outlined below.

The EDGAR and GPW proxy datasets that are used for most of the proxy data are initially processed at the highest resolution available (*e.g.*, 0.1° for the EDGAR emissions data) and split into countries, including also a split into land and ocean areas, at this resolution before being aggregated to the final 0.5° resolution used for the data products.
Because we allow multiple countries to exist within one 0.5° grid cell, this results in a more accurate distribution of emissions for each country, even when data is being processed at the lower 0.5° resolution. For the few datasets with a resolution lower than 0.5°, data are sub-sampled to a resolution of 0.5° using an appropriate template.

### 2.3 Open Burning Emissions

The development of historical open burning emissions is described in van Marle et al. (2017). In brief, the spatial
emissions distribution for these emissions is fundamentally grid-based, with emissions over the satellite era estimated from remotely sensed data (GFED4s, 1997-2015) merged with several existing historical proxies for fires including charcoal records, for boreal and temperate North America and Europe, and visibility-based fire emissions, for the tropical areas of Equatorial Asia and the Arc of Deforestation. The spatial distribution for the pre-satellite era was

based on the 1997-2015 average and uniformly adjusted for large geographic regions based on the proxies (see van Marle et al., 2017 for the basis-regions used). For the regions where proxies had limited coverage the output of six different fire models from the Fire Model Intercomparison Project (FireMIP) were used (Rabin et al. 2017).[1] Data is reported from 1750-2015. Before 1997 the annual regional fire emissions were distributed over a 0.25 × 0.25 degree grid based on the GFED4s climatology for six classes (savanna and grassland fires, deforestation fires, boreal forest fires, temperate forest fires, peat fires and agricultural waste burning).

Future open burning emissions originate from the integrated assessment models (IAMs), which report by model region and broad category (e.g. forest or grassland burning). While many of the models have some additional level of spatial detail, emissions were reported at the model region level to facilitate common data harmonization, downscaling, and gridding routines. Future open burning emissions, therefore, were mapped to spatial grids using the same methodologies as used for anthropogenic emissions, as further described below. This means that, unlike historical open burning emissions, the spatial distribution of open burning emissions within a category (e.g. forest burning) within any country does not vary in the future scenarios.

## 2.4 Future Emissions

Scenario data for future emissions from IAMs are first harmonized to a common 2015 base year value by native model region and sector. This harmonization process adjusts the native model data to match the 2015 starting year values with a smooth transition forward in time, generally converging to native model results (Gidden et al. 2018). The production of the harmonized future emissions data is described in Gidden et al. (2019).

The 2015 base-year anthropogenic emissions data by country and sector are extensions of the 2014 historical data largely using emission factor trends for combustion sources from the GAINS model (ECLIPSE V5a; Stohl et al., 2015, Klimont et al. 2017) and BP fuel consumption statistics (BP 2016). Non-combustion sources were generally scaled by estimated population. There are potentially large changes in emissions over this period, for example in China (Zheng et al. 2018), which results in uncertainty of these estimates for regions with rapid changes in air pollution control technology deployment. Near-term $SO_2$ emission trajectories in China were adjusted to better match the estimates of (Zheng et al. 2018) during the harmonization process.

For open burning, the 2015 base-year emissions data for harmonization and downscaling are an average of the previous 10 years of historical data extracted by country. This is because inter-annual variability is not captured in the future projections, so a longer-term average is a more appropriate starting point for open burning emissions. While a decadal averaging period was used here, a longer averaging period could also be considered in future work.

The global integrated models (IAMs) that generated the SSP emission projections each have different number of socio-economic regions (11-32) for which the unharmonized emission data are available. Harmonization to the common 2015 starting dataset, therefore, occurs at each individual model's native spatial definition in order to preserve as much detail from the model as possible (Gidden et al. 2019). Harmonized emissions for the native model regions, are then downscaled to the country level as described in Gidden et al. (2019). The country-level downscaling is performed in order to provide a uniform basis for subsequent mapping to spatial grids but does not necessarily represent specific policies that might be in place for any particular country.

After the downscaling procedure the country-level future emissions projections are at the same level of sectoral resolution as the final gridding sectors in Table 2. These are then mapped to a spatial grid using the same underlying methodology as for the historical anthropogenic data, with a few differences in detail. Spatial proxies for each country are taken to be the 2014 gridded historical emissions discussed above for anthropogenic emissions and the average of the last 10 years (2005-2014) of historical data from van Marle et al. (2017) for open burning emissions. These spatial proxies are, therefore, constant into the future and do not represent any shifts in the spatial location of emissions within a country. The one exception is international shipping, which uses year-specific spatial distributions from the ECLIPSE project for the years 2015, 2020, 2030, 2040, and 2050. These distributions capture projected changes in shipping fuel sulfur content and imposition of low sulfur control areas near coastal areas.

Future open burning emissions are provided from the IAMs in the following categories: Agricultural Waste burning on Fields (AWB), forest burning, grassland burning, and peatland burning. Future trends for these emissions are generated by each model driven by, in large part, changes in land use. Note that these are future projections of

---

[1] https://www.imk-ifu.kit.edu/firemip.php

climatologically average emission rates over time and do not include inter-annual variability. Because the future model projections used here do not include data on peatland burning, peatland burning emissions are held constant into the future.

Future emissions were gridded at the aggregate sectoral level that corresponds to the final gridding sectors in the historical emissions data for anthropogenic emissions (Table 2), and with forest burning combined into one category, aggregating the open burning historical categories of: boreal forest fires, temperate forest fires, and tropical deforestation and degradation. This lower level of sectoral detail was used because the models that generated the future scenario data used in this process (Gidden et al. 2018) often lack the finer level of sector detail that was available in the historical emissions datasets. The future data was constructed so that the data were consistent at the grid-cell level when moving from the historical (up to 2014) to the future (2015 and forward) dataset.

The second major difference is that future emissions are not produced annually, but are provided for 2015, 2020, and at decadal intervals thereafter. This is because long-term models do not provide annual data. The choice was made to only distribute data for years that originally was produced by the models, and not interpolated data. This also means that fewer, and smaller, data files need to be downloaded and processed by end-users. We note that the format of the open burning emissions in the future data is slightly different than the format used in the historical data. This is because we use the same software for both open burning and anthropogenic future emissions, so a similar data format was used for all future emissions data.

### 2.4.1 Future Net Negative Fossil $CO_2$ Emissions

In the default historical and future emissions gridding, emissions from electric power generation and some other large industrial facilates are generally mapped to a spatial grid as large point sources. For most emission species this is an appropriate representation of emissions currently and also into the future. For future carbon dioxide emissions, however, an inconsistency occurs for net negative $CO_2$ emissions if this method were applied. Net negative $CO_2$ emissions occur in models when biofuel feedstocks such as crop residues, wood, or dedicated biomass crops are used to produce energy (e.g., electricity or liquid biofuels) and the $CO_2$ that is produced as part of either transformation or combustion processes is then sequestered in geologic reservoirs. The models report these net negative emissions at the point of conversion or combustion, that is, at the power plant or bio-refinery. Physically, however, carbon dioxide is taken out of the atmosphere at the point where the biofuel is produced, generally in agricultural lands. The $CO_2$ uptake in scenarios with net negative $CO_2$ emissions, therefore, would actually occur over a wide area and not at a point source. Representing net negative $CO_2$ emissions as a point uptake is not only physically incorrect but could cause Earth system model codes to error by causing very low or negative $CO_2$ concentration values.

We, therefore, distribute net negative $CO_2$ emissions over a global spatial grid. Our procedure is to first downscale harmonized model emissions to the country level (Gidden et al. 2019), and then sum all net negative $CO_2$ emissions. Any remaining positive $CO_2$ emissions are gridded normally by country. The global sum of net negative $CO_2$ emissions are then distributed spatially using a global map of potential biofuel productivity. This map was derived by taking the potential production from the LPJ-ml land model as included in the IMAGE model (Müller et al. 2016). The IMAGE-LPJ model generates potential production maps (in dry matter tons) for several crops taking into consideration the climatic and soil conditions for that specific grid cell. We have used the maximum yield of woody and grassy bio-energy crops for each 0.5 x 0.5 grid cell, resulting a global map of potential bio-energy production. Only those cells allocated as 2010 agricultural area (including bio-energy production) have been used. Global total net negative $CO_2$ emissions are then spatially allocated at each time step following this bio-energy production map. The spatial allocation distribution is, therefore, constant in time, although the absolute magnitude of total net negative $CO_2$ emissions varies with time and scenario (Gidden et al. 2019).

A global allocation was chosen because biomass trade information was not available from the models. While a more detailed time-changing map could be produced, carbon-cycle modelers indicated that a global distribution would be sufficient for use in future scenarios. Note that net negative $CO_2$ emissions are allocated to a new gridded emissions sector, which means that the future $CO_2$ emission files have one more element in the sector dimension than the historical $CO_2$ emission files.

Note that this procedure is only applied for net-negative $CO_2$ emissions. Combustion of biomass-based fuels in the absence of CCS is not included in the $CO_2$ emissions data. Any net change in $CO_2$ emissions due to use of biofuels is accounted for in either net land-use change emissions (Gidden et al. 2019) or as part of changing land-use and land cover (Hurtt et al. 2011).

## 2.5 Seasonality

Monthly seasonality is applied to the gridded emissions by applying a set of spatially-explicit seasonality fractions. The primary source for the seasonality fractions was the ECLIPSE dataset, with the addition of EDGAR v4.3 for international shipping and Lamarque et al. (2010) for aircraft. The dataset was processed to a consistent seasonality matched the monthly calendar used in the final emissions data (*e.g.*, a 365 day year). As part of this process some minor inconsistencies in the anthropogenic seasonality data between sectors in terms of number of days assumed in the year were corrected so that the month distribution was consistent with the 365 day year used in the anthropogenic dataset. Seasonality for future open burning emissions was taken from a ten year average from the historical dataset.

We note that, after distribution of this dataset, it was found that the industrial sector in some regions has a high level of seasonality in the ECLIPSE dataset, and this distribution was carried through to the CMIP6 data (As noted on the on-line GitHub issues list for this dataset, see below). In general, we would expect industrial sector emissions to be fairly constant as most industrial activities operate year-round. The magnitude of seasonality overall in some regions, therefore, may be overestimated. This will be re-examined in future data releases.

## 2.6 VOC Speciation

Modelling atmospheric chemistry requires emissions for specific species, or species groups, of reactive compounds instead of the total mass of volatile hydrocarbon emissions (NMVOCs) generally tabulated in inventories. Anthropogenic VOC emissions were speciated by applying speciation profiles by gridding sector and country, largely derived from the RETRO project. Emissions for 23 anthropogenic VOC compound groups were extracted by country and broad sector from gridded HTAP v2 emissions data.[2] These were normalized and converted to percentages and applied to the emissions data at the level of country and final gridding sector. In the future, it would be useful to use country and sector-specific profiles, such as those in Huang et al. (2017). The speciated emissions were gridded in the same manner as bulk VOC emissions thereby providing emission grids by species.

While speciation profiles by gridding sector are held constant in time, the aggregate VOC speciation for a country will change over time. This is because of different speciation profiles for each sector combined with a changing sectoral contribution to total VOC emissions over time.

Figure 2 shows the ratio of Butanes to Hexanes+ in the RETRO dataset used in CEDS and the newer estimate by Huang et al. (2017), which we present as an illustration of differences in speciation profiles used in these two datasets. While there is considerable overlap between the two datasets, the RETRO data has a lower average ratio for these two species in the road transportation sector as compared to EDGAR while the opposite is the case for industrial combustion. We note that variation across, and even within, countries is expected. For example, Chin et al. (2012) report that the fraction of n-Heptane (a higher Alkane) in gasoline vapors varies between 6-19%, depending on location and season within the United States.

For both historical and future emissions, however, VOC speciation profiles can potentially change with technology deployment separately from changes in NMVOC emission rates. For example, regulations to limit ozone formation not only result in altered bulk NMVOC emission amounts, but also change the specific VOC species emitted (Kirchstetter et al. 1999). Other examples include changes over time in the composition of consumer products (McDonald et. al 2018) and changing formulation of paints and other coating materials.

It is not known how much the differences illustrated in Figure 2 reflect updated or different data sources, or actual changes in VOC speciation over time. The RETRO dataset is targeted at 1990/1995 and uses speciation profiles available in time period, while the EDGAR data takes in newer measurements. While the emission time series presented here, and also the EDGAR data, both capture the broad changes in VOC speciation due to changing sectoral contributions to VOC emissions over time, these datasets do not capture underlying changes in speciation profiles due to changing regulations or other fundamental changes to speciation profiles over time. It would be useful, therefore, to determine the importance of such speciation changes for historical modeling of atmospheric chemistry, as compared to the broader changes in VOC emission magnitudes and speciation changes due to sectoral shifts over time. Note that the Huang et al. (2017) data do capture changes in speciation over time due to changing mix of fuels in each end-use sector, which is illustrated in Figure 2.

---

[2] http://iek8wikis.iek.fz-juelich.de/HTAPWiki/WP1.1

For future open burning emissions, speciation profiles by country and open burning sector were extracted from the historical open burning emissions data files and applied to the future scenario bulk NMVOC emissions. For details on the historical open burning VOC speciation see van Marle et al. (2017). Note that the species provided are different for the anthropogenic emissions and open burning emissions, following past conventions used by each of those communities.

**3 Results**

In this section we provide a brief overview of the gridded data products, focusing largely on diagnostic graphics that illustrate the underlying structure and potential issues that impact use and interpretation of the gridded data.

*3.1 Gridded products*

Table 4 provides a summary of the gridded data files described in this paper with emission species listed in Table 1. The files listed there, plus historical gridded open burning emissions (van Marle et al. 2017) provide the complete set of emissions data required for CMIP6 experiments (Eyring et al. 2016). As discussed in Hoesly et al. (2018) the anthropogenic emissions are consistently generated from the same driver data across sectors and emission species. The gridded data files are generated using consistent spatial proxy data, seasonality, and sector definitions. This means that some emission species, such as $CO_2$, which have been provided as only bulk emissions in CMIP5, are provided here with greater sectoral detail. For applications that do not require this level of sectoral detail, emissions can be summed across sectors. Note that, for all species, total anthropogenic emissions consist of the sum of all eight (historical) or nine (future)[3] surface emission sectors plus aircraft emissions.

Supplementary data files include speciated NMVOC emissions for all time periods. Also provided are historical emissions from solid biomass fuels (wood, agricultural residues, etc.) as these are used as supplementary data in some models.[4] The solid biofuel emissions are a sub-set of the total emissions in the main data files and are provided for models that need this information. Solid biofuel emissions were not reported separately for the future scenarios so these are not provided in the gridded data. If these emissions are needed for the future scenarios, we recommend that the ratio of solid biofuel to total emissions by sector/grid cell from the last few historical years times the future emissions data for each species be used to derive future grids for solid biofuel emissions in the future scenarios. More detailed emissions data by sector and fuel from individual models could be used to access the adequacy of this assumption.

Spatial distributions for anthropogenic reactive gases and carbonaceous aerosol emissions (e.g., BC and OC) for selected historical and future years are shown in Figures 3-6, with numerical values in Table 3. An overview of large-scale trends is provided here, with the emission trajectories and their derivation described in more detail in the underlying literature for the historical emissions (van Marle et al. 2017, Hosley et al. 2018), future scenarios (Calvin et al. 2017, Fricko et al. 2017, Fujimori et al. 2017, Kriegler et al. 2017, Riahi et al. 2017, van Vuuren et al. 2017) and the harmonized CMIP6 data (Gidden et al. 2019).

Global emissions of species that are the product of incomplete combustion (BC, CO, NOx, OC, NMVOC) are dominated by open burning in 1850 (Table 3, Figures 3-4). Anthropogenic combustion emissions in 1850 are dominated by solid biofuel use for cooking and heating. However, by the mid-20th century, anthropogenic processes dominate for all species except for OC (Table 3). As industrialization proceeds, emissions become large in the regions that experienced early industrialization such as Europe, North America, Japan, and Australia by the mid to late 20th century (e.g., Figure 5). By the early 21st century (e.g. 2015, Figure 6) emissions controls have decreased pollutant emissions somewhat in Europe and North America but increased dramatically compared to mid-20th century levels in the emerging economy regions such as China, India, and the Middle East. Anthropogenic $CO_2$ emissions, in general, do not follow the same trajectory as air pollutant emissions.

Future emissions diverge depending on the emission scenario and follow trends as described in Gidden et al. (2019). A few illustrative graphics are provided here. By 2050 in the SSP2, "middle of the road" scenario, emissions have

---

[3]  As discussed above, future emissions contain an additional sector to represent net negative $CO_2$ emissions.
[4]  Some models use information on solid biofuel combustion to derive information on associated VOC emissions and/or primary aerosol (e.g., BC, OC) characteristics.

decreased from 2015 levels in most world regions (Figure 7), while decrease to quite low levels in the SSP1-26 scenario (Figure S7). In contrast, emissions do not decrease nearly as much in the SSP3-70 scenario (Figure S9). By 2100, under the SSP2-45 scenario, anthropogenic are generally larger than open-burning emissions, but anthropogenic + open burning emissions are generally smaller than 2015 levels. Future open-burning emissions also
vary by scenario, but not as dramatically as anthropogenic emissions (Figures S3-6).

### 3.2 Gridded Data Diagnostics and Interpretation

While 0.5 degree gridded data is the primary product of gridding, maps at that resolution can be of limited use for understanding spatial and temporal patterns in the data or diagnosing potential issues. In this section we discuss a number of diagnostic graphs that illustrate various dimensions of the dataset. We first discuss the temporal structure
of the anthropogenic emissions data over different sectoral and spatial scales.

**Figure 8**a shows a time series plot of $SO_2$ emissions at one grid cell, for the industrial (IND) gridding sector (e.g. combustion + process emissions, see Table 2). These emissions are the result of a combination of changes in the proxy data over time, in this case from EDGAR emissions, multiplied by the estimated time series for the United States for the underlying detailed sectors. Emissions from this same cell for the Energy sector (**Figure 8**b) show a
sharp increase, due to (presumably) a power plant coming into service, as represented in the proxy dataset.

Total anthropogenic $SO_2$ emissions for the same grid cell (**Figure 8**c) show a different temporal structure, as a result of combing the differing temporal trends from different sectors. Emissions become smoother if we examine total $SO_2$ emissions summed over 16 nearby grid cells (**Figure 8**d). However, emissions at either the grid cell or for a sum of several grid cells will generally show more structure over time as compared to the corresponding emissions for the
entire USA region (**Figure 9**) due to the additional temporal structure in the proxy dataset at smaller spatial resolutions.

For any particular spatial location, the accuracy of the gridded emissions depends on the accuracy of the underlying proxy dataset as well as underlying the country-level emissions data. Global proxy datasets that are available for use in project such as this, including projects of similar scope such as EDGAR (from which much of the spatial
distributions used here are drawn), will not necessarily correspond exactly to the actual spatial location and magnitude of emission sources. For example, databases of global road networks, while improving (e.g., Crippa et al 2018), will not exactly correspond to either actual road traffic at every point, nor are they likely complete for all regions. While large emission sources, such as power plants, are more likely to be in global databases, such data can still be incomplete, particularly for smaller plants (Liu et al. 2015). For other sectors, it is difficult to capture in proxy
datasets historical changes over time in emission strengths in specific spatial locations.

At the grid cell level, therefore, discontinuities over time can potentially be due to: a) discontinuities in the country level emission time series, b) actual changes in the underlying emitting processes as represented in the proxy dataset, or c) data breaks or inconsistencies in the proxy dataset that do not represent an actual historical change in emissions. As either the sectoral aggregation or the spatial scale increases, however, the data will tend to become more robust, at
least to the extent the country-level emission time series used for calibration in this project are accurate (see Hoesly et al. 2018 for further discussion).

Monthly time series plots visualizing the seasonal cycle of the emissions data, as shown in **Figure 10**, are also useful s diagnostics. Note, again, that discontinuities at the single cell level are due to temporal structure in the proxy datasets.

The temporal signature of the open burning data is quite different, as inter-annual variations in those data are dominated in large part by year-to-year changes in local meteorological conditions. At least for the modern era, the inter-annual variations are inferred from satellite observations. As discussed below, increased use of remote sensing and regional bottom-up data has the potential to improve the spatial and temporal accuracy of anthropogenic emissions data as well.

Spatial data presented at a lower resolution can also be useful in understanding overall patterns, and spatial differences between datasets. As discussed in Hoesly et al. (2018), we have found that spatial maps at a 10° resolution are useful in showing large-scale difference in emissions magnitude and spatial distribution between two datasets. For anthropogenic $SO_2$ emissions in 1900, for example (e.g., Figure 11) the CMIP6 dataset shows some

shifts in emissions from north-west to Eastern Europe and a different distribution of emissions over the oceans compared to the CMIP5 dataset.

Note also that the CMIP6 dataset has a different distribution of $SO_2$ emissions over North America than in the CMIP5 data, with more $SO_2$ emissions over the western areas, and lower emissions over the east. This has been
recognized (Yang et al. 2018) as a bias in the proxy dataset over the United States used here (e.g. EDGAR $SO_2$ emissions). As discussed above, national level anthropogenic emissions were distributed within each country using proxy data. In this case, the proxy data likely does not account for the lower sulfur content of coal in the western United States. As discussed below, one method of accounting for such issues is to estimate emission at smaller geographic units, such as US states, and then map to spatial grids. For further comparisons between the CMIP5 and
CMIP6 anthropogenic emissions data see Hoesly et al. (2018).

**4 Discussion and Potential Future Work**

The spatially distributed emissions data discussed here represent a number of improvements over previous century-scale gridded datasets. Anthropogenic emissions were consistently generated from the same driver data, mapped to spatial grids using the same proxy data across emission species and have consistent seasonality. Future emissions are
downscaled to the country level and then consistently mapped to spatial grids using largely the same gridding methodologies as used for the historical anthropogenic emissions.

Emission estimates remain uncertain owing to data gaps, biases and errors in activity data, bias in emission factors, and spatial and temporal variations in real-world conditions. Relative uncertainties will generally increase going back in time as less detailed sectoral, spatial, and temporal information is available. Future emissions projections are
inherently uncertain, which is why multiple scenarios are presented. Within any one scenario, no attempt was made in these data products to estimate how the spatial distributions of emissions within countries might change into the future. It is not clear, however, the practical value of attempting such detail given the overall uncertainty in future trajectories.

The historical open burning emissions have specific sources of uncertainties, as discussed further in van Marle et al.
(2017). These start with uncertainties of the different products used in the construction of the overall dataset: GFED4s, emissions stemming from visibility observations, charcoal-based fire records and modelled emissions. The choice for which fire models are used before the satellite observed fires (1997-2015) can influence the results, as will the land-use change data and other assumptions used to drive those models. The use of GFED4s climatology to distribute emissions spatially and temporally (e.g. seasonally) from regional estimates before 1997 assumes that sub-
regional distributions stayed constant before 1997.

While any errors in aggregate emission estimates will flow down to errors in spatial distribution, here, however, we are particularly interested in potential errors in allocating emissions spatially. Systematic analysis of uncertainties in spatial emission patterns, however, have not been assessed to date. The methodologies used here for historical anthropogenic and future scenario emissions rely on an underlying spatial proxy data. One source of error is when the
spatial proxy data does not match the actual spatial distribution of emissions. One example was discussed above in the case of $SO_2$ emissions in the United States. In this case, the global proxy dataset used to distribute these emissions spatially did not contain country-specific information on the sub-regional distribution of emissions. Overall, as the emissions data are considered over larger spatial areas, we expect that accuracy is likely to improve.

Capturing the distinctions between urban and rural emissions, and finer distinctions in-between, is an on-going challenge for emission inventories, for example road transport emissions, open refuse burning, and fuel combustion for residential heating/cooking. Driving conditions and emission profiles tend to differ between urban and rural conditions, resulting in different fuel consumption and emission factors. To the extent regional inventories become better able to capture these distinctions, those methodologies could ultimately be incorporated into the methods for generating proxy data for global datasets.

While the focus here was the use of consistent spatial proxy data across species, in some cases it might be desirable to use varying proxies between pollutants, e.g. coal use in residential sector will produce significant $SO_2$ while heating or cooking with wood will not. Some of these differences, however, may be captured in the EDGAR emissions distributions used as proxy data. Because country-level anthropogenic emissions estimates were used for historical emissions gridding, potential mis-allocation of emissions is most important for countries with large spatial extents.

More general spatial errors can also occur because of miss-matches between proxy data and actual emission sources. Traffic volume, for example, is not likely to scale purely with the class of roadway in global road datasets. Country-level emissions from power plants, for example, are allocated spatially using proxy data such as power plant capacity and the presence of emission control devices, which are the primary information available in global proxy datasets. Actual emissions, however, won't necessarily be proportional to facility capacity, resulting in spatial errors. Further, point sources can be missing in global proxy datasets, particularly in developing countries (Liu et al. 2015). In addition, the spatial location of individual sources might not always accurate in proxy datasets.

Additional kinds of error can occur in the temporal dimension. Even if the time series at the aggregate country level is correct, emissions at any spatial location will likely not follow the same temporal pathway. The emissions rate of localized sources is likely to change over time, and individual point sources, could close down, or new sources put into operation. Temporal consistency over time at the grid-cell level is perhaps one of the most difficult challenges for long-term gridded datasets. Improved information on emissions over time for large point sources (see below) is the most important issue here, as year-to-year changes in large sources could be a large source of temporal changes in emissions.

The distribution of emissions over the year, represented in these datasets as monthly average seasonality, is another area where improvement is likely needed. The seasonal distribution for anthropogenic emissions is based on global sectoral analysis, for example mapped by heating-degree days (HDDs), that is not likely to incorporate country-specific details in how HDDs map to actual fuel use. The rate of heating fuel consumption relative to temperature will likely be different in a cold region of a country (where heating demand is high) as compared to a region with a milder climate using different heating technologies. The seasonal distribution of ammonia emissions depends on sector and region (Paulot 2014) and it is not clear if there is a general consensus on these seasonal distributions.

There are several potential strategies for reducing such biases and errors in gridded datasets. One general approach is to improve the completeness and representativeness (relative to actual emissions) of global proxy datasets, to reduce the issues discussed above. One example is community efforts to improve global databases, such as those for power plants and road networks, used in emissions work.

Another method is to estimate emissions at smaller geographical units, such as states or provinces and allocate emissions spatially at that level. Emissions information is often available at sub-national levels, for example for US states (US EPA), China provinces (Liu et al. 2015), and Indian states (Klimont et al. 2017). Allocating emissions information at this level would result in an improved overall distribution of emissions over large countries, even if the same global proxy data-sets were used for each individual state or province. For global-scale models this would likely be sufficient accuracy, assuming the state-level emission time series were accurate. An effort is underway, for example, to implement a state level disaggregation for the USA in the CEDS, which would fix the East-West $SO_2$ emissions distribution issue mentioned above.

Detailed spatial data for emissions have been developed over many regions for air quality modelling purposes, and it is likely that incorporating these data into global inventories could result in improved spatial data overall. One method is to simply "stitch" together detailed spatial maps where they are available (Janssens-Maenhout et al. 2015). One disadvantage of this approach, however, is that the resulting datasets are often inconsistent because detailed spatial data is not available for every year; different years in different regions need to be used. Inconsistencies also arise at country borders, where grid cells overlap. Another disadvantage is that these detailed data are often time

consuming to generate, which means that these regional datasets are not available for the most recent years. An alternative to binding together emission data grids is to incorporate the underlying spatial proxy distributions into workflows such as those described here. This would, presumably, result in improved spatial distributions for those regions with data, but also be flexible enough to allow greater temporal consistency between regions. One challenge,

however, is that these detailed datasets are often only available for recent decades. For longer-term modelling, some method of extending these proxy data back into time would be needed, perhaps by blending with simpler proxy data such as population distributions.

For large point sources in some regions direct measurements of emissions through Continuous Emission Monitoring (CEM) systems are in place, which potentially provide highly accurate emission estimates for those specific sources.

Another source of emission information for large sources is satellite remote sensing data. Fioletov et al. (2016) for example, have used satellite data to develop a catalogue of emission sources "missing" in existing gridded inventories. Either of these data sources could be combined with bottom up estimates to produce a more robust emission dataset, with the spatial location and temporal (e.g. year-to-year) variations of emissions from large sources provided by CEMs or remote sensing analysis. This was done, in a limited manner, for $SO_2$ emissions from

petroleum refining in the CMIP6 dataset (Hoesly et. al. 2018).[5] Note, however, there are limits to such techniques in that satellite data is only reliably estimated for the largest sources. Spatial and temporal biases in remote sensing data need to be carefully considered, for example, the greater difficulty of detecting sources in winter as compared to summer.

Additional data sources on specific large sources, such as literature sources and corporate reports, can also be

valuable for providing temporal emissions detail. While such information was used in the CEDS CMIP6 release, for example for $SO_2$ emissions from several large ore smelting complexes,[6] this was only applied at the country level, not at the level of point sources. Information on industrial sources is particularly difficult to obtain given the wide diversity of source types and, often, confidentiality issues preventing data availability for specific facilities. Given that large point sources often account for a large portion of emissions for some species, adding a general capability to

represent specific point sources and their emission levels over time, as distinct from more numerous smaller sources, would improve the accuracy of emissions data.

While there are many aspects of emissions data that can potentially be improved, is will be important to better assess which improvements are most critical for focused efforts. For example, is a 10% error in overall emissions level in a region more or less important than a 20% East-West error in spatial allocation? These tradeoffs will depend on the

model and specific application. For example, smaller-scale details will be less important for coarser-scale models. It would be useful to more systematically test the sensitivity of models to spatial allocation. While localized results will depend on spatial distribution, it is not clear how sensitive larger-scale results, such as circulation, temperature, and precipitation changes, are to differences in spatial allocation. One work directly examining this sensitivity (Geng et al. 2018), found that, once power plant point source and road network proxy data were used, there was a significant

benefit from using industrial economic activity data to spatially allocate industrial sector emissions.

The current datasets produced for CMIP6 were produced at either 0.5° (anthropogenic) and 0.25° (open burning over 1750-2015). Given that primary purpose of these datasets is for global models, this resolution is appropriate given that this is finer than the spatial resolution of most current global models, particularly those used to conduct century-scale simulations. Higher resolution datasets would be useful for regional modelling or next-generation models with

models with higher resolution. In principle, given the availability of 0.1° EDGAR emissions data used as proxy data within CEDS, emissions data up to this level of resolution are feasible to produce. It would be useful, before committing to production of long-term higher resolution emissions data to examine the accuracy of the global proxy datasets at this level of spatial detail (and also the accuracy required for a specific purpose). This could be done, for example, by comparing to higher resolution regional data.

Emission datasets would benefit also from more systematic comparisons with observations. Some observations, such as concentration ratios, can be directly compared to inventories (e.g. Hoesly et al. 2018), while, more generally, modelled concentrations need to be compared to observations. Where model biases can be ruled out, differences with

---

[5] https://GitHub.com/JGCRI/CEDS/wiki/Data_and_Assumptions#so-sub-2-sub-1

[6] For example: the Kazakhmys smelter in Kazakhstan, Raahe sintering plant in Finland, and the Algoma sintering plant in Canada. See the CEDS on-line documentation (footnote 5) for details.

observations can be used to inform improvements to inventory data. At present, however, this is performed in an ad hoc manner, with no regular, systematic comparison across species and regions.

Finally, it would be useful if the types of long-term emission datasets discussed here could be more regularly updated. In the past, most of these datasets are compiled once every 5 or so years in order to produce data for the next CMIP exercise. A paradigm of continuous improvement would allow more incremental releases, allow new emissions data to be more thoroughly tested, and result in fewer differences between each subsequent dataset release. This is already the case for the GFED open burning data for the satellite era, and is in progress for the CEDS anthropogenic data.

## 5 Data Availability

The codes and data described in this paper are available in a number of open source data and code repositories.

Historical Anthropogenic Emissions: https://GitHub.com/JGCRI/CEDS
*Methodological details and user guide at: https://GitHub.com/JGCRI/CEDS/wiki*
*Data issues are documented in the "issues" section at this same location.*

Historical Open Burning Emissions: http://www.globalfiredata.org/ar6historic.html

Future Emissions Downscaling and Gridding: https://GitHub.com/iiasa/emissions_downscaling
*Methodological details and user guide at: https://GitHub.com/iiasa/emissions_downscaling /wiki*

CMIP6 Gridded Data: https://esgf-node.llnl.gov/search/input4mips/

## Author Contributions

LF developed historical and future emissions gridding methodology and code, produced historical anthropogenic gridded emissions, and developed initial paper outline and text. SJS administered project, participated in data development, and produced initial full paper text. CB extended future emissions gridding code and produced future scenario gridded emissions. LF, MC, MG, ZK MvM, MvdB, and GRvdW commented on paper text. MC, ZK, MvdB, and RH supplied key data for anthropogenic emissions. MvM and GRvdW produced data on historical open burning emissions. MG led the overall production of the harmonized future emissions scenarios.

The authors declare that they have no conflict of interest.

## Acknowledgements

This research at JGCRI was support by the National Aeronautics and Space Administration's Atmospheric Composition: Modeling and Analysis Program (ACMAP), award NNH15AZ64I (historical emissions gridding), the U.S. Department of Energy, Office of Science, as part of research in Multi-Sector Dynamics, Earth and Environmental System Modeling Program (future scenario gridding), and the DOE Office of Science, Biological and Environmental Research (historical emissions development). The Pacific Northwest National Laboratory is operated for DOE by Battelle Memorial Institute under contract DE-AC05-76RLO1830. Work on future scenario data received funding from the European Union's Horizon

2020 research and innovation programme under grant agreement # 641816 (CRESCENDO). The authors thank Kalyn Dorheim for helpful comments.

**A.1 Gridded Data Release History**

5    We provide below a summary of the release history for gridded emissions data.

**Historical Anthropogenic Emissions**

- June and July 2016: preindustrial (1750-1850) data (ver. 2016-06-18) and historical 1851-2014 (ver. v2016-07-26) for aerosol and reactive gas species were published on ESGF.

- September 2016: files were re-published in a new format (with sectors contained within a dimension) due to a limitation in ESGF software on some nodes. (version numbers were amended with -sectordim) Data values did not change.

- May 2017: New data version (ver. 2017-05-18) that corrected some errors in the previous gridded data (ver. 2016-06-18, and 2016-07-26, see README and analysis of difference between the new and old data sets at: https://goo.gl/tGrc27). The new release also included CO2 emissions (all years) and CH4 emissions (from 1970 forward). Note that aggregate country emissions by sector did not change, only the spatial distribution within each country.

- August 2017: "Rough cut" extension of CH4 emissions between1850 and 1970 (in decadal increments) for models that require CH4 emissions over all years. Labeled as supplemental-data (ver. 2017-05-18) on ESGF.

- September 2017: New versions of aircraft gridded emissions (ver. 2017-08-30 and 2017-10-05 for SO2) were published on ESGF. This is a cosmetic change only, whereby emissions were set to zero before 1920.

**Historical Open Burning Emissions**

- V1.2 (December 2016). Sectoral contributions for all species are now included (in v1.1 only for CO). Minor modifications to the interannual variability between 1960 and 1997 for tropical regions.

- V1.1 (November 2016). Updated emissions, mainly for the boreal regions. Global total fire carbon emissions decreased by about 5% (~100Tg C yr-1) over 1750-2015.

- V1.0 (~September 2016). First release of all emission species including NMVOC speciation.

**Future Emissions**

- As of this writing only one version of the future emissions has been published. Most species and data files were published on ESGF on 2018-06-28. Fossil CO2 emissions and H2 emissions from open burning were published on 2018-12-19.

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

## Tables and Figures

| Emissions Data Set | Emission Species Provided |
|---|---|
| Anthropogenic Bulk Emissions | $CO$, $CH_4$, $NH_3$, $NO_X$, $SO_2$, NMVOC, BC, OC, $CO_2$ |
| Open Burning Bulk Emissions | $CO$, $CH_4$, $NH_3$, $NO_X$, $SO_2$, NMVOC, BC, OC, $H_2$ |
| Speciated Anthropogenic Emissions | alcohols, ethane, propane, butanes, pentanes, hexanes_plus_higher_alkanes, ethene, propene, ethyne, other_alkenes_and_alkynes, benzene, toluene, xylene, trimethylbenzenes, other_aromatics, esters |
| Speciated Open Burning Emissions | C10H16, C2H2, C2H4, C2H4O, C2H5OH, C2H6, C2H6S, C3H6, C3H6O, C3H8, C5H8, C6H6, C7H8, C8H10, CH2O, CH3COCHO, CH3COOH, CH3OH, HCN, HCOOH, Higher-Alkanes, Higher-Alkenes, HOCH2CHO, MEK, Toluene-lump |

**Table 1.** Emission species provided in the emission data files. For molecular weights assumed for each species see: https://GitHub.com/iiasa/emissions_downscaling/wiki.

| Final gridding sector | Intermediate gridding sector definition | Proxy Data Source | Years |
|---|---|---|---|
| **Historical Anthropogenic Emissions** | | | |
| Residential, commercial, other (RCO) | Residential, Commercial, Other (Residential and Commercial) | HYDE Population (interpolated annually) | 1750 - 1899 |
| | | EDGAR v4.2 (1970) blended with HYDE | 1900 - 1969 |
| | | EDGAR v4.2 RCORC | 1970 – 2008 |
| | Residential, Commercial, Other (Other) | HYDE Population (interpolated annually) | 1750 - 1899 |
| | | EDGAR v4.2 (1970) blended with HYDE | 1900 - 1969 |
| | | EDGAR v4.2 RCOO | 1970 – 2008 |
| Agriculture (AGR) | Agriculture | EDGAR v4.2 AGR | 1970 – 2008 |
| Energy sector (ENE) | Electricity and heat production | EDGAR v4.2 ELEC | 1970 – 2008 |
| | Fossil Fuel Fires | EDGAR v4.2 FFFI | 1970 – 2008 |
| | Fuel Production and Transformation | EDGAR v4.2 ETRN | 1970 – 2008 |
| | Oil and Gas Fugitive/Flaring | ECLIPSE FLR 1990, 2000, 2010 EDGAR v4.2 ETRN (1970 - 2008) | 1970 – 2010 |
| Industrial sector (IND) | Industrial Combustion | EDGAR v4.2 INDC | 1970 – 2008 |
| | Industrial process and product use | EDGAR v4.2 INPU | 1970 – 2008 |
| Transportation section (TRA) | Road Transportation | EDGAR v4.3.2 ROAD (2010) * | 1750 – 2014 |
| | Non-road Transportation | EDGAR v4.2 NRTR | 1970 – 2008 |
| International shipping (SHP) | International Shipping | ECLIPSE + additional data (1990 – 2015) | 1990 - 2010 |
| | Tanker Loading | ECLIPSE + additional data (1990 – 2015) | 1990 - 2010 |
| Solvents (SLV) | Solvents production and application | EDGAR v4.2 SLV | 1970 – 2008 |
| Waste (WST) | Waste | HYDE Population, GPW v3 (modified rural population) | 1750 – 2014 |
| Aircraft (AIR) | Aircraft | CMIP5 (Lamarque et al., 2010; Lee et al., 2009) * | 1850 – 2008 |
| **Additional Sector for Future Anthropogenic Emissions** | | | |
| Negative $CO_2$ (NEGCO2) | Net Negative $CO_2$ Emissions | See main text * | 2015 – 2100 |
| **Future Open Burning Emission Sectors** | | | |
| Forests | Forest fires, including deforestation | 2005-2014 average (van Marle et al. 2017) | 2015 – 2100 |
| Grassland | Grassland fires | 2005-2014 average (van Marle et al. 2017) | 2015 – 2100 |
| Peat | Peatland fires | 2005-2014 average (van Marle et al. 2017) | 2015 – 2100 |
| Ag Waste (AWB | Agricultural Waste Burning on fields | 2005-2014 average (van Marle et al. 2017) | 2015 – 2100 |

**Table 2** Proxy Data used for gridding anthropogenic emissions data, adapted from Hoesly et al. (2018). Gridding proxies marked with an asterisk (*) do not vary in time. Future emissions use the same final gridding sectors, with the addition of four sectors for open burning emissions (*e.g.,* Agricultural Waste burning on Fields (AWB), forest burning, grassland burning, and peatland burning), and an additional

10   sector for net Negative $CO_2$ Emissions (NEGCO2). A decadal average (2005-2014 ave) for open burning proxy data was used for future emissions as indicated. Anthropogenic sector definitions can be found in Hoesly et al. (2018). Note that, except for international shipping, all future emissions used proxy data at the level of country and sector that did not change over time, see main text.

| Species | Source | 1850 | 1980 | 2015 | 2050 | 2100 | Units |
|---|---|---|---|---|---|---|---|
| BC | anthro | 934 | 4,975 | 7,968 | 4,769 | 1,415 | |
| | open-burn | 1,644 | 1,646 | 1,748 | 1,548 | 1,211 | ktC |
| CH4 | anthro | 31,306 | 271,364 | 373,737 | 342,274 | 280,322 | |
| | open-burn | 11,919 | 12,086 | 14,336 | 14,893 | 14,830 | ktCH4 |
| CO | anthro | 118,527 | 593,946 | 606,222 | 445,376 | 162,521 | |
| | open-burn | 294,472 | 297,994 | 327,549 | 284,852 | 220,688 | ktCO |
| NH3 | anthro | 6,028 | 42,442 | 61,383 | 71,465 | 60,414 | |
| | open-burn | 3,599 | 3,560 | 3,894 | 4,564 | 4,953 | ktNH3 |
| NMVOC | anthro | 10,782 | 137,690 | 163,887 | 143,351 | 80,164 | |
| | open-burn | 56,315 | 57,025 | 63,270 | 53,646 | 40,272 | ktNMVOC |
| NOx | anthro | 558 | 99,289 | 138,653 | 98,457 | 56,610 | |
| | open-burn | 19,838 | 20,353 | 21,060 | 18,146 | 13,493 | ktNO2 |
| OC | anthro | 4,262 | 11,313 | 19,538 | 13,441 | 3,823 | |
| | open-burn | 14,045 | 14,008 | 15,202 | 13,417 | 10,702 | ktC |
| SO2 | anthro | 2,481 | 135,141 | 98,326 | 50,649 | 28,077 | |
| | open-burn | 2,073 | 2,090 | 2,162 | 2,170 | 2,031 | ktSO2 |
| CO2 | anthro | 188 | 20,331 | 11,464 | 22,585 | 31,396 | MtCO2 |

**Table 3** Total anthropogenic and open burning global emissions for selected time periods. Values for 2050 and 2100 are from the SSP2-45 scenario. Only anthropogenic $CO_2$ emissions are shown since net open burning $CO_2$ emissions depend on land-use change history and are not provided in this dataset.

| Data Description | Year Range | Example Filename on ESGF |
|---|---|---|
| Anthropogenic Emissions - Surface | Historical (1750-2014) | CO-em-anthro_input4MIPs_emissions_CMIP_CEDS-2017-05-18_gn_185101-189912.nc |
| Anthropogenic Emissions - Aviation | Historical (1750-2014) | CO-em-AIR-anthro_input4MIPs_emissions_CMIP_CEDS-2017-10-05_gn_185001-185012.nc |
| Speciated Anthropogenic Emissions | Historical (1750-2014) | VOC01-alcohols-em-speciated-VOC-anthro_input4MIPs_emissions_CMIP_CEDS-2017-05-18-supplemental-data_gn_180001-184912.nc |
| Solid Biofuel Emissions | Historical (1750-2014) | OC-em-SOLID-BIOFUEL-anthro_input4MIPs_emissions_CMIP_CEDS-2017-05-18-supplemental-data_gn_195001-199912.nc |
| Extended Anthropogenic $CH_4$ Emissions | Historical (1750-2014) | CH4-em-anthro_input4MIPs_emissions_CMIP_CEDS-2017-05-18-supplemental-data_gn_185001-196012.nc |
| Anthropogenic Emissions - Surface | Future (2015-2100) | CO-em-anthro_input4MIPs_emissions_ScenarioMIP_IAMC-MESSAGE-GLOBIOM-ssp245-1-1_gn_201501-210012.nc |
| Anthropogenic Emissions - Aviation | Future (2015-2100) | CO2-em-AIR-anthro_input4MIPs_emissions_AerChemMIP_IAMC-AIM-ssp370-lowNTCF-1-1_gn_201501-210012.nc |
| Speciated Anthropogenic Emissions | Future (2015-2100) | VOC07-ethene-em-speciated-VOC-anthro_input4MIPs_emissions_ScenarioMIP_IAMC-IMAGE-ssp126-1-1-supplemental-data_gn_201501-210012.nc |
| Open Burning Emissions | Future (2015-2100) | NOx-em-anthro_input4MIPs_emissions_ScenarioMIP_IAMC-IMAGE-ssp119-1-1_gn_201501-210012.nc |
| Open Burning Sector Shares | Future (2015-2100) | CO-openburning-share_input4MIPs_emissions_ScenarioMIP_IAMC-IMAGE-ssp119-1-1_gn_201501-210012.nc |
| Speciated Open Burning Emissions | Future (2015-2100) | NMVOC-C6H6-em-speciated-VOC-openburning_input4MIPs_emissions_ScenarioMIP_IAMC-REMIND-MAGPIE-ssp534-over-1-1-supplemental-data_gn_201501-210012.nc |
| Speciated Open Burning Sector Shares | Future (2015-2100) | NMVOC-CH2O-speciated-VOC-openburning-share_input4MIPs_emissions_ScenarioMIP_IAMC-IMAGE-ssp126-1-1-supplemental-data_gn_201501-210012.nc |

[*]Historical emission files are generally provided in 50-year segments in order to keep the size of individual files reasonable (historical $CH_4$ emissions are provided over 1970-2014. Supplementary emissions files provide an approximate extension back to 1850). Data for the CMIP6 starting year 1850 are provided in one file. Future emissions were provided for the entire projection period (2015-2100) in one file per species/emissions type enumerated in this table. All data except aviation are provided as two-dimensional grids, generally with an additional sector dimension. Aviation emissions are provided as three-dimensional data. All data available at: https://esgf-node.llnl.gov/projects/input4mips/.

**Table 4.** Gridded emission data files provided for CMIP6. The complete historical anthropogenic dataset consists of 327 files. The complete future scenarios dataset consists of 1017 files. Historical emissions are provided annually, while future emissions are provided for 2015, 2020, and in decadal intervals after that. All emission are provided with monthly seasonality.

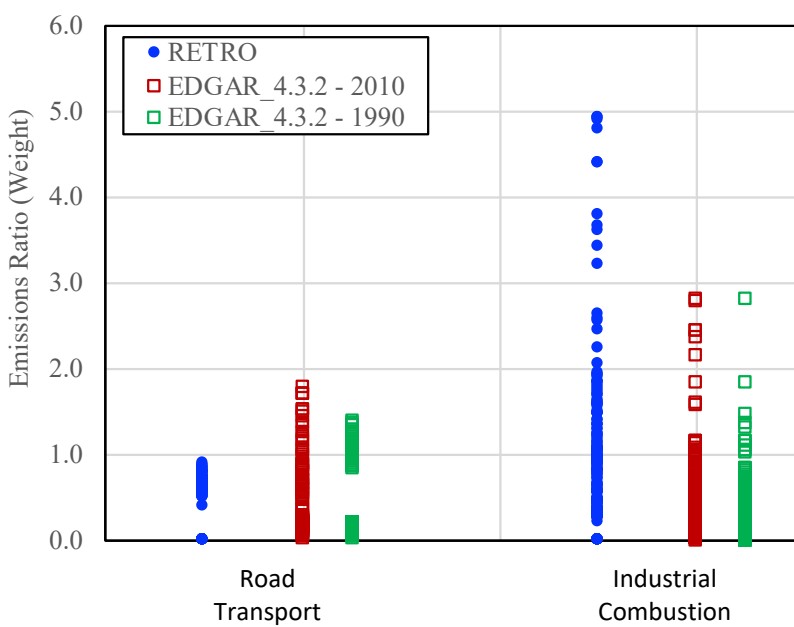

**Figure 2** – Range across countries of the Butane/Hexanes (and higher alkanes) in the RETRO speciation used in CEDS and the results from Huang et al. 2017) for 2010 and 1990. Each symbol represents one country. A lower ratio indicates more of the higher weight alkanes in the assumed speciation profile. The difference between the 1990 and 2010 EDGAR data are due to a changing mix of fuels used for those sectors between those two time periods, for example shifts between different amount of gasoline vs diesel fuel consumption.

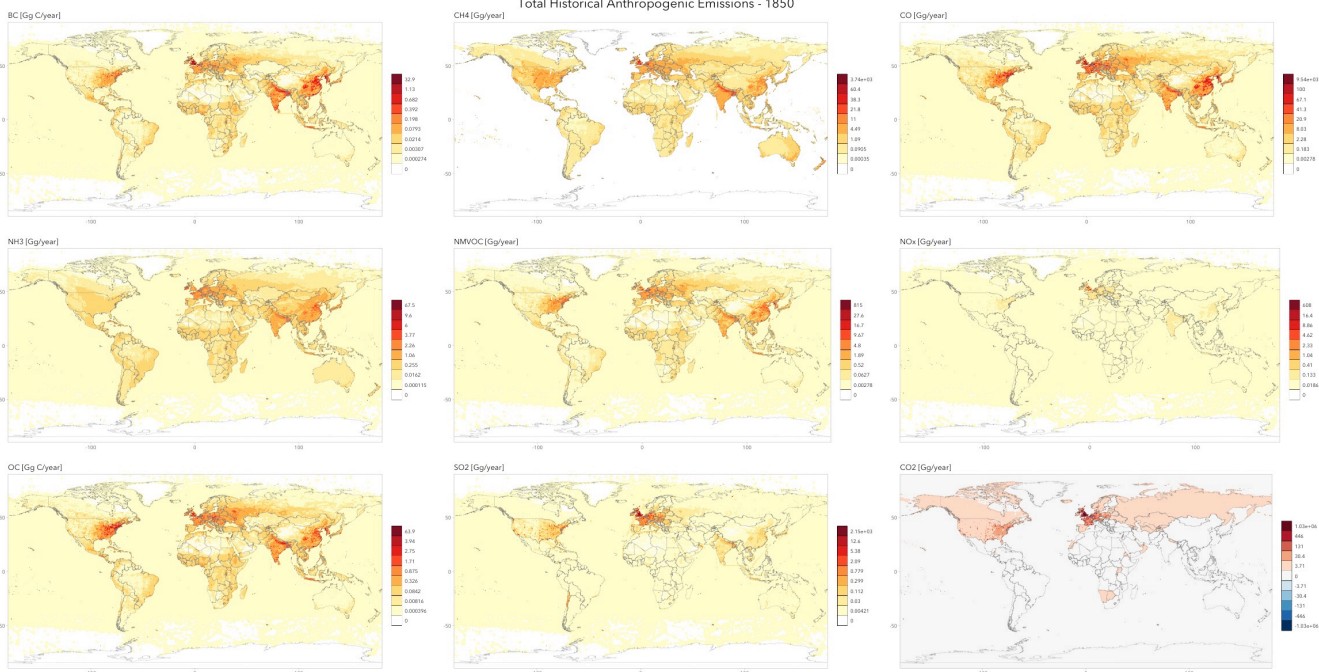

**Figure 3** – Total anthropogenic emissions in 1850 by species. Color scale is the same for each emission species across all gridded figures for comparison. Note that the $CO_2$ color scale accommodates the net negative emissions found in some future scenarios.

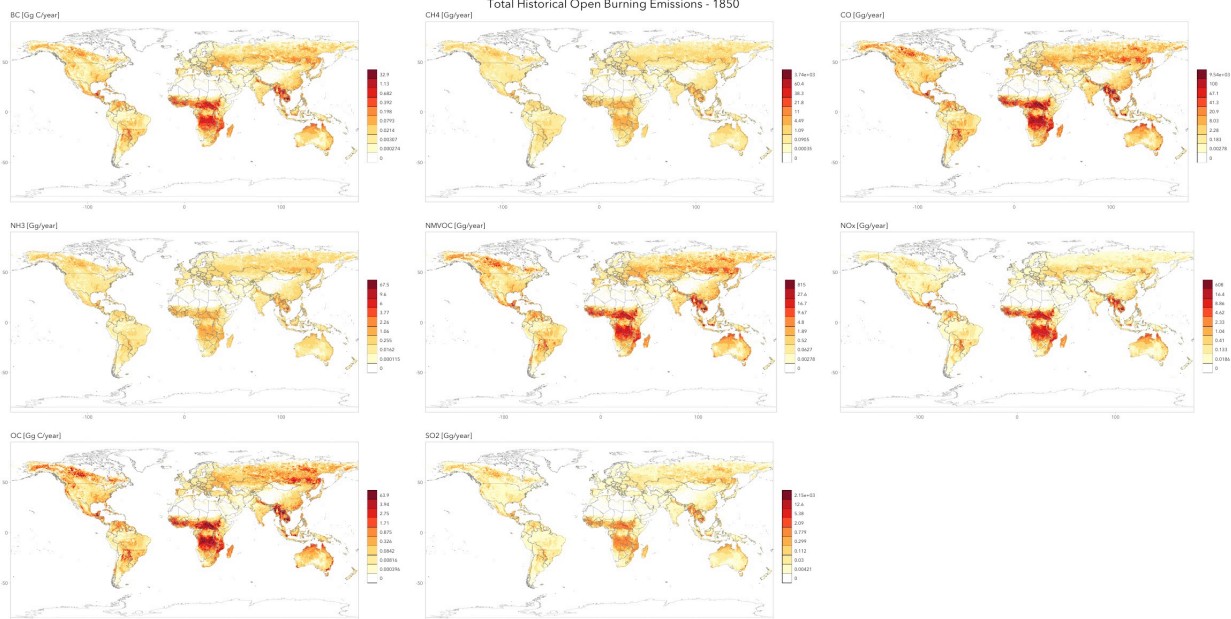

**Figure 4** – Total open burning emissions in 1850 by species. Color scale is the same for each emission species across all gridded figures for comparison.

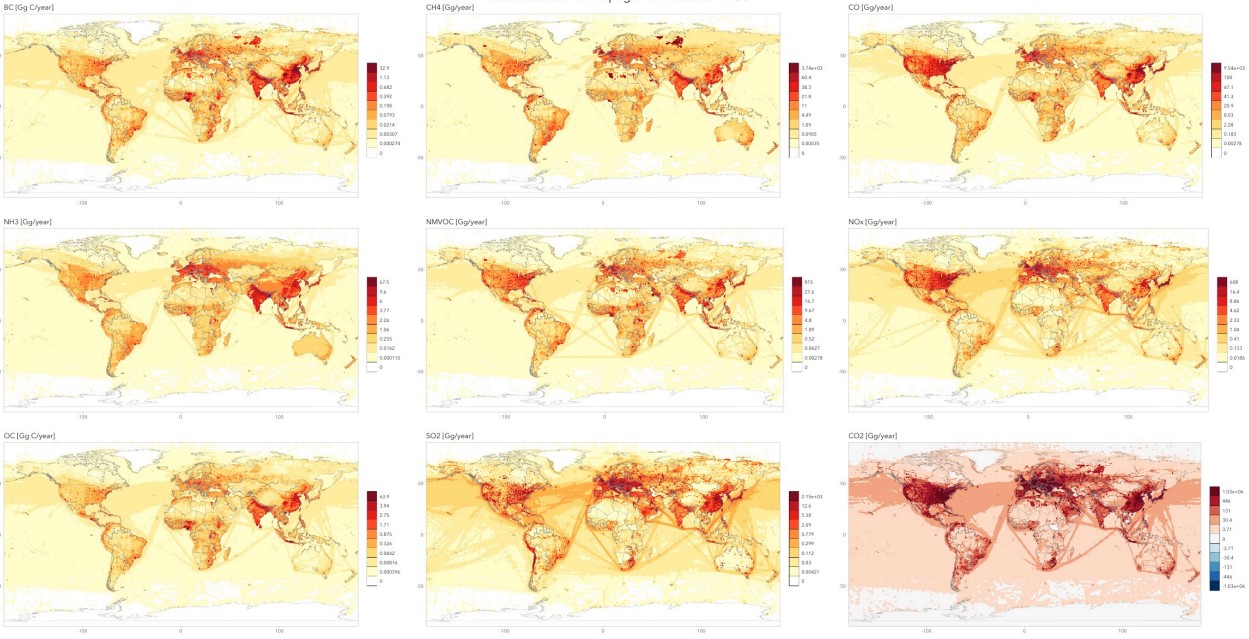

5    **Figure 5** – Total anthropogenic emissions in 1980 by species. Color scale is the same for each emission species across all gridded figures for comparison. Note that the $CO_2$ color scale accommodates the net negative emissions found in some future scenarios.

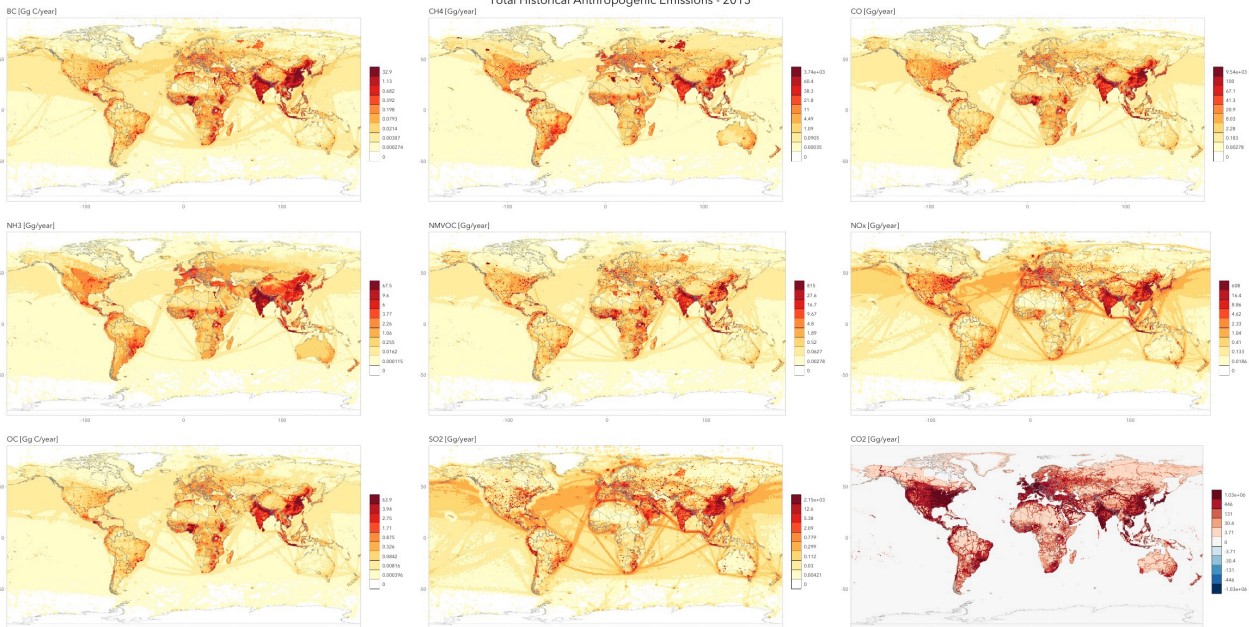

**Figure 6** – Total anthropogenic emissions in 2015 by species. Color scale is the same for each emission species across all gridded figures for comparison. The $CO_2$ color scale accommodates net negative emissions found in some future scenarios.

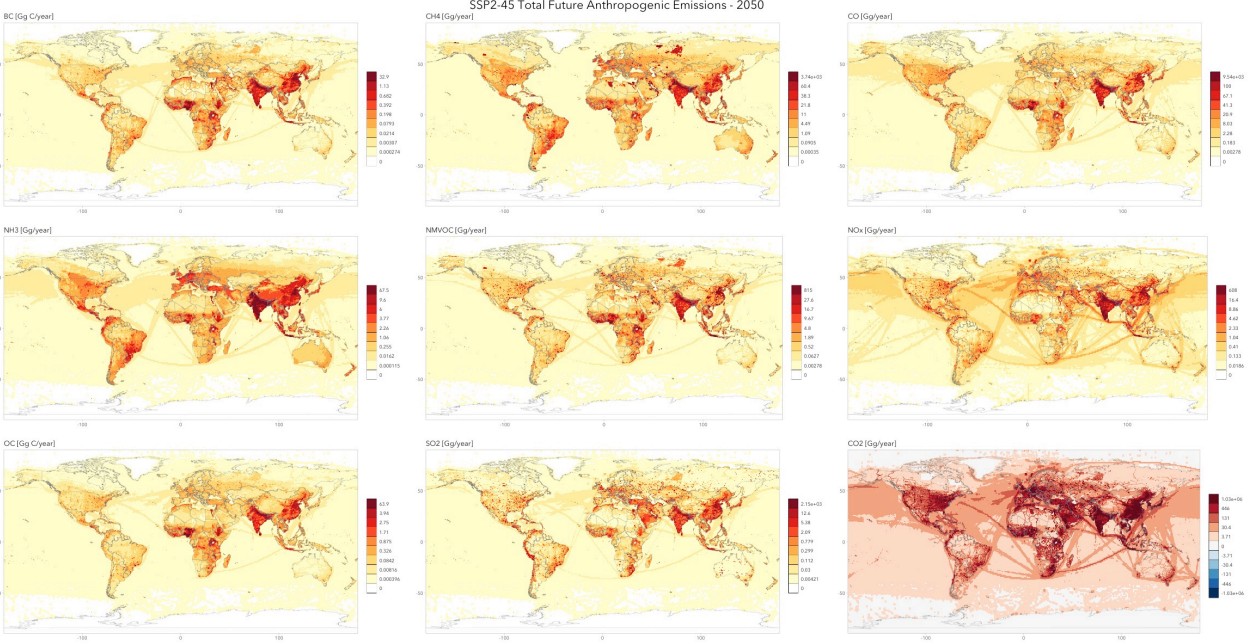

**Figure 7** – Total anthropogenic emissions in 2050 by species in the SSP2-45 scenario. Color scale is the same for each emission species across all gridded figures for comparison. The $CO_2$ color scale accommodates net negative emissions found in some future scenarios.

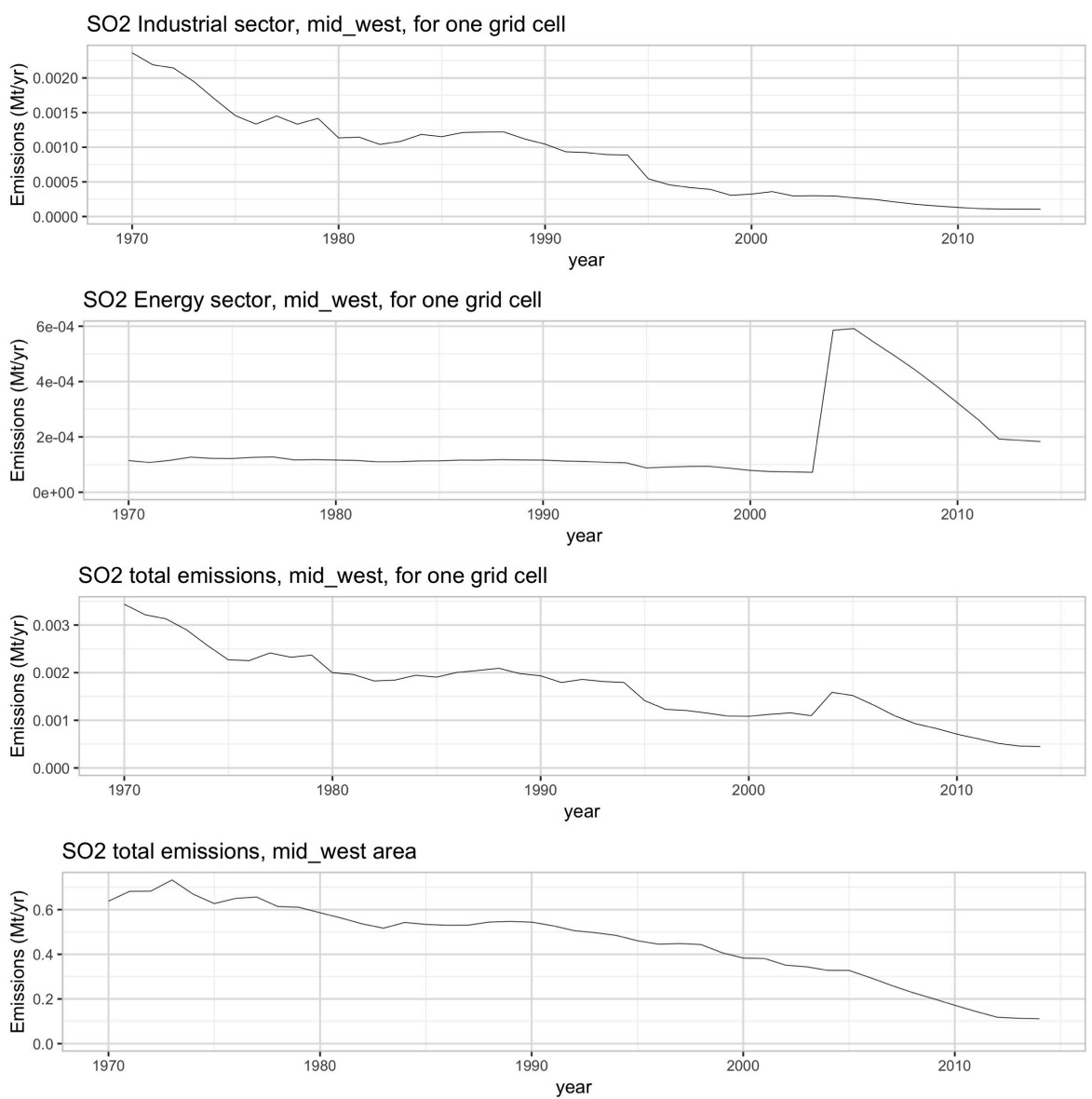

**Figure 8** – a) Time series plot for the industrial gridding sector for a single 0.5 degree cell on the US mid west (lat, lon: 41.25, -87.75). b) Time series plot for the Energy sector for the same grid cell. c) Time series plot for total anthropogenic emissions for the same grid cell. d) Time series plot for total

10    anthropogenic emissions summed over 16 grid cells including the cell depicted in a-c.

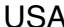

USA

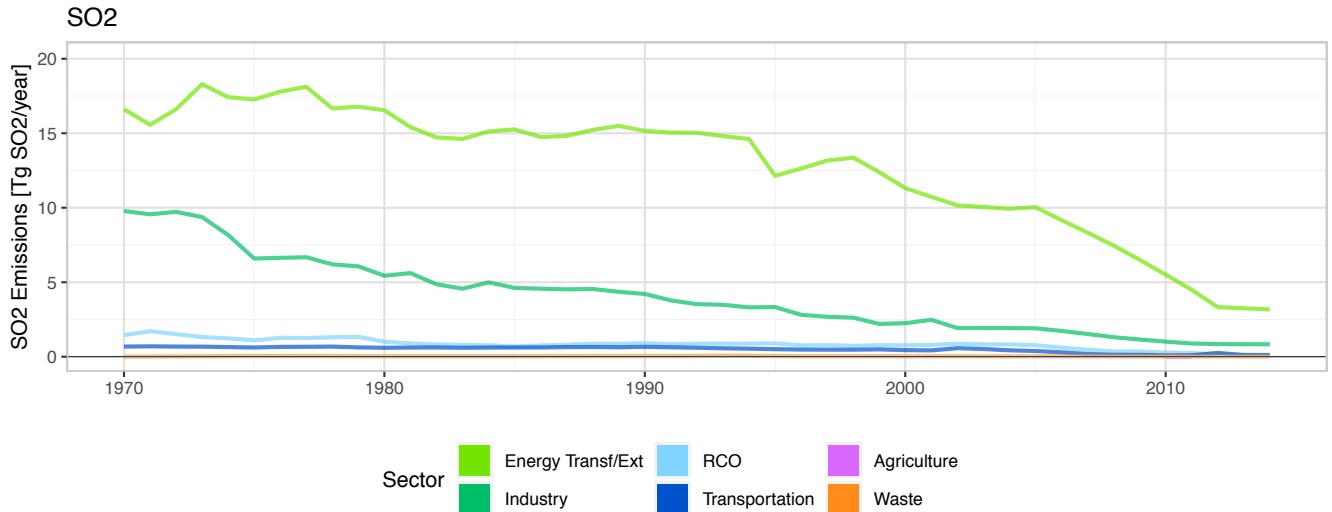

**Figure 9** – Time series plot for SO2 by sector for the USA as a whole.

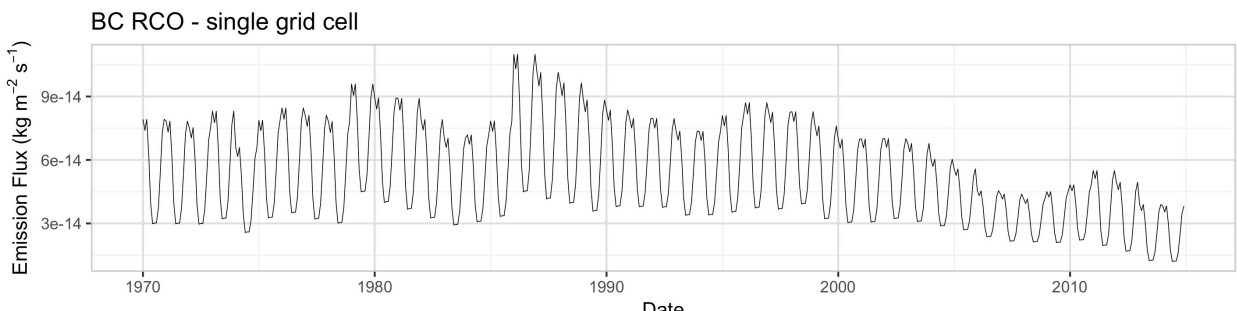

5 **Figure 10** – Time series plot for total BC emissions over time at the monthly level for a grid cell in Finland.

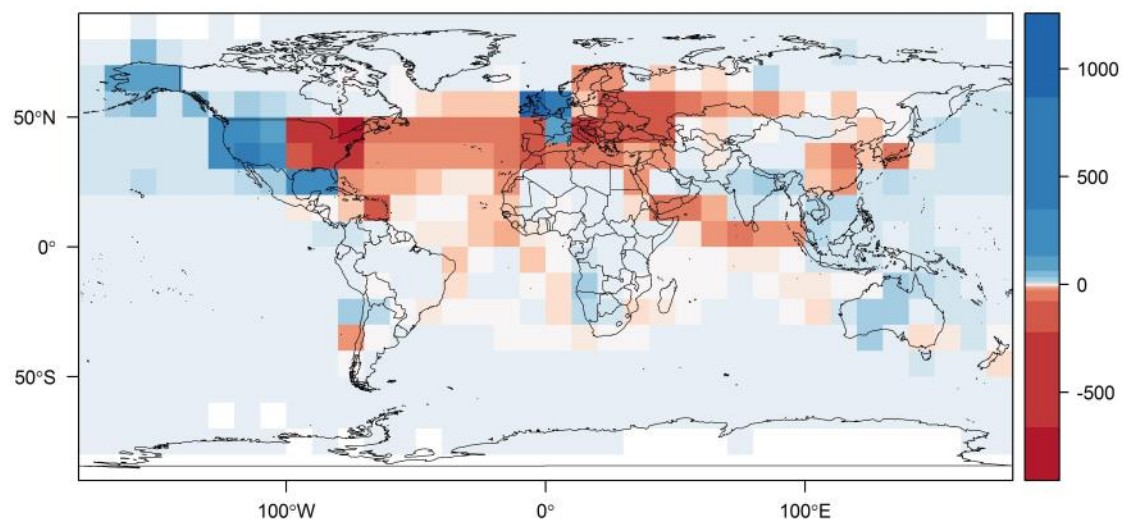

**Figure 11** – Difference in CMIP6 (e.g. CEDS) and CMIP5 gridded total $SO_2$ emissions in 1900. For like
10 with like comparison, these figures do not include aviation, or agricultural waste burning on fields.