# Peer review of "The Generation of Gridded Emissions Data for CMIP6"

_Geoscientific Model Development, 2019_

## Referee Comment (RC1) · Anonymous Referee #1 · 5 Sep 2019

This manuscript documents the methods used for generating gridded emission inventory particularly for the CMIP6. The authors described the automated framework that has been developed along with a brief diagnostics and interpretation of the products. In the end, they also discussed the uncertainties, limitations, and potential future work. I believe this paper is suitable for publication after addressing the following comments.

General comments:

(1) This manuscript documents the methods used for generating gridded data sets, instead of describing the CMIP6 emission product itself (which is the focus of Hoesly et al. [2018]). The title of this manuscript "Gridded Emissions for CMIP6" may not match the content and focus of this manuscript.

(2) Section 2 ("Data and Methodology") has only one sub-section 2.1 ("Methodology overview"). Since there's no sub-section 2.2, it seems unnecessary to have the subsection 2.1. I assume the "Data" in the title "Data and Methodology" indicates the input data such as GFED and EDGAR that are described in the "Methodology overview", as there is no separate sub-section 2.2 for Data.

(3) The main text and the figures seem to be de-coupled. Many of the figures (such as Figures 8-10) are only mentioned but not discussed in detail in the main text. I'm wondering if these figures could be explained more, or moved to Supplementary Material.

(4) Section 4 Discussion and Potential Future Work are very long. Perhaps the authors could have sub-sections (e.g., 4.1 Spatial errors, 4.2 Temporal errors, 4.3 potential strategies and future directions . . .) for this Section.

(5) The discussion on the uncertainties in Section 4 appears to be qualitative. If possible, please provide more numbers and/or references for quantified or estimated uncertainties.

Minor comments:

(1) Page 1 Line 17. "The development of . . . future scenarios were coordinated". Please change "were" to "was".

(2) Page 2 Lines 24-25. "As discussed below, the future gridded emissions builds on these two historical datasets and, in large part, inherits their properties such as within-country spatial distribution and seasonality." Please change "builds" and "inherits" to "build" and "inherit", respectively.

(3) Page 6 Lines 27-28. "carbon-cycle modelers indicated that a global distribution would be sufficient for use in future scenarios." Please provide reference(s) for this statement.

(4) Page 6 Lines 32. Please provide a full name for CCS.

(5) Page 7 Lines 16-19. Figures S3-7 and S9 are mentioned here. However, I can't find the Supplementary Material for this manuscript. Please provide it.

(6) Section 3.2. Please describe how the chosen grid cell is selected and if this chosen grid in the US is representative.

(7) Page 11 Lines 27. ", is will be important to better assess . . .". I believe you have a typo in this sentence.

(8) Table 1. In the third row, the Emission Species Provided are shown by the names of the species, while in the fourth row, the Emission Species Provided are shown by chemical formulae. Please use consistent representation in this Table.

(9) Figures 2-6. The font in the Figures is too small to be seen. Please adjust it.
* * *

---

## Referee Comment (RC2) · Anonymous Referee #2 · 17 Sep 2019

This paper document the gridded emissions for CMIP6. The description is in very detail and useful for the users. I suggest publishing the paper after minor revision.

General comments:

1. Page 4, line 10. The authors used gridded emissions as proxy data. It is easy to understand such usage if the gridded emissions share the same spatial resolution with the needed proxy. What if the spatial resolutions are not the same?

2. Section 2.1.4. The gridded HTAP v2 emissions data is used as proxy to speciate VOC. I'm wondering how reliable this speciation algorithm is if the HTAP emissions differ significantly from the CMIP6 emissions. The authors pointed out that it would be useful to use country and sector-specific profiles, such as those in Huang et al. (2017). I recommend selecting a small domain to perform an intercomparsion for speciated VOC derived from both algorithms. This will contributes to a better understanding of

the uncertainty of VOC speciation in this study.

3. Conclusion. "The spatially distributed emissions data discussed here represent a number of improvements over previous century scale gridded datasets." I recommend a short summary of the improvements compared to previous approaches here.

Specific comments:

1. Figure 1. What is the reason for using multiple boxes to represent "Spatially distributed emissions for each country"?

2. Figure 2. The legend is too small to identify.

---

## Editor Comment (EC1) · Jason Williams (Editor) · 26 Nov 2019

Dear Authors,

Before the manuscript can be accepted in its final form can you address the following issues:

(i) Missing software name and version identifier. Software is provided for generating the data sets. This software needs to be named and versioned, and the name and version included in the paper title. This is important as the methods and software presented may change in the future and it is important that others are able to clearly identify the version that this manuscript describes. (ii) Github URLs. Github is an excellent development platform, but it lacks the features required of an archive. GitHub themselves tell authors to use Zenodo for this purpose. The authors should follow the procedure detailed here to archive the exact version of the software used to create the results

presented: https://guides.github.com/activities/citable-code/ . The resulting Zenodo repositories present the correct bibliography entries to use. (iii) A website is cited for historical burning emissions, this is insufficiently persistent. Instead (or as well), the corresponding paper should be cited. I believe it is: https://doi.org/10.5194/gmd-10-3329-2017. (iv) The reference to the CMIP6 data from the code and data section is too generic to be at all useful. This is only a technical issue, though, because it appears that the exact identifiers are in table 4. Table 4 should be referenced from the code and data availability section (this may seem petty, but readers of GMD papers know to go to this section to find the data). (v) In table 4 itself filenames are given to identify data on ESGF. This is unsafe. The ESGF records themselves provide the correct DOIs to cite, and if the authors click through those DOIs they get the full citation and even corresponding BibTeX and RIS files. The filenames should be replaced with proper data citations.

Thank you for your attention towards these issues.

Jason Williams.

---

## Author Comment (AC2) · 7 Jan 2020

The author response has been revised to include responses to editorial and technical comments.

Please also note the supplement to this comment:
https://www.geosci-model-dev-discuss.net/gmd-2019-195/gmd-2019-195-AC2-supplement.pdf

---

## Author Response (AR2)

**The Generation of Gridded Emissions Data for CMIP6**

**Response To Reviewers**

We thank the reviewers for their helpful comments that raised useful points to add to the manuscript. Reviewer comments and questions are marked in blue, with our response in black text below

Anonymous Referee #1

This manuscript documents the methods used for generating gridded emission inventory particularly for the CMIP6. The authors described the automated framework that has been developed along with a brief diagnostics and interpretation of the products. In the end, they also discussed the uncertainties, limitations, and potential future work. I believe this paper is suitable for publication after addressing the following comments.
General comments:
(1) This manuscript documents the methods used for generating gridded data sets, instead of describing the CMIP6 emission product itself (which is the focus of Hoesly et al. [2018]). The title of this manuscript "Gridded Emissions for CMIP6" may not match the content and focus of this manuscript.

We thank the referee for their comment. We have revised the title to be:

**The Generation of Gridded Emissions Data for CMIP6**

which better clarifies the focus of this manuscript.

(2) Section 2 ("Data and Methodology") has only one sub-section 2.1 ("Methodology overview"). Since there's no sub-section 2.2, it seems unnecessary to have the sub-

We have clarified the header numbering, and added a header for the discussion of historical emissions gridding.

Anonymous Referee #2

This paper document the gridded emissions for CMIP6. The description is in very detail and useful for the users. I suggest publishing the paper after minor revision.
General comments:
1. Page 4, line 10. The authors used gridded emissions as proxy data. It is easy to understand such usage if the gridded emissions share the same spatial resolution with the needed proxy. What if the spatial resolutions are not the same?

This is an excellent point. We have added the following text to describe this part of the processing in greater detail. (We note that the actual software is used for the historical anthropogenic and future emissions is all open source.)

The EDGAR and GPW proxy datasets that are used for most of the proxy data are initially processed at the highest resolution available (*e.g.*, 0.1° for the EDGAR emissions data) and split into countries, including also a split into land and ocean areas, at this resolution before being aggregated to the final 0.5° resolution used for the data products. Because we allow multiple countries to exist within one 0.5° grid cell, this results in a more accurate distribution of emissions for each country. For the few datasets with a resolution lower than 0.5°, data are sub-sampled to a resolution of 0.5° using an appropriate template.

2. Section 2.1.4. The gridded HTAP v2 emissions data is used as proxy to speciate VOC. I'm wondering how reliable this speciation algorithm is if the HTAP emissions differ significantly from the CMIP6 emissions. The authors pointed out that it would be useful to use country and sector-specific profiles, such as those in Huang et al. (2017). I recommend selecting a small domain to perform an intercomparsion for speciated VOC derived from both algorithms. This will contributes to a better understanding of C1

This is an excellent suggestion. We have conducted a comparison and added the following discussion and figure to the paper.

While speciation profiles by gridding sector are held constant in time, the aggregate VOC speciation for a country will change over time. This is because of different speciation profiles for each sector combined with a changing sectoral contribution to total VOC emissions over time.

Figure 2 shows the ratio of Butanes to Hexanes+ in the RETRO dataset used in CEDS and the newer estimate by Huang et al. (2017), which we present as an illustration of differences in speciation profiles used in these two datasets. While there is considerable overlap between the two datasets, the RETRO data has a lower average ratio for these two species in the road transportation sector as compared to EDGAR while the opposite is the case for industrial combustion. We note that variation across, and even within, countries is expected. For example, Chin et al. (2012) report that the fraction of n-Heptane (a higher Alkane) in gasoline vapors varies between 6-19%, depending on location and season within the United States.

For both historical and future emissions, however, VOC speciation profiles can potentially change with technology deployment separately from changes in NMVOC emission rates. For example, regulations to limit ozone formation not only result in altered bulk NMVOC emission amounts, but also change the specific VOC species emitted (Kirchstetter et al. 1999). Other examples include changes over time in the composition of consumer products (McDonald et. al 2018) and changing formulation of paints and other coating materials.

It is not known how much the differences illustrated in Figure 2 reflect updated or different data sources, or actual changes in VOC speciation over time. The RETRO dataset is targeted at 1990/1995 and uses speciation profiles available in time period, while the EDGAR data takes in newer measurements. While the emission time series presented here, and also the EDGAR data, both capture the broad changes in VOC speciation due to changing sectoral contributions to VOC emissions over time, these datasets do not capture underlying changes in speciation profiles due to changing regulations or other fundamental changes to speciation profiles over time. It would be useful, therefore, to determine the importance of such speciation changes for historical modeling of atmospheric chemistry, as compared to the broader changes in VOC emission magnitudes and speciation changes due to sectoral shifts over time. Note that the Huang et al. (2017) data do capture changes in speciation over time due to changing mix of fuels in each end-use sector, which is illustrated in Figure 2.

[Figure]

**Figure 2** – Range across countries of the Butane/Hexanes (and higher alkanes) in the RETRO speciation used in CEDS and the results from Huang et al. 2017) for 2010 and 1990. Each symbol represents one country. A lower ratio indicates more of the higher weight alkanes in the assumed speciation profile. The difference between the 1990 and 2010 EDGAR data are due to a changing mix of fuels used for those sectors between those two time periods, for example shifts between different amount of gasoline vs diesel fuel consumption.

3. Conclusion. "The spatially distributed emissions data discussed here represent a number of improvements over previous century scale gridded datasets." I recommend a short summary of the improvements compared to previous approaches here.

The following two sentences in the text summarize the major improvements, namely:

*Anthropogenic emissions were consistently generated from the same driver data, mapped to spatial grids using the same proxy data across emission species and have consistent seasonality. Future emissions are downscaled to the country level and then consistently mapped to spatial grids using largely the same gridding methodologies as used for the historical anthropogenic emissions.*

We have edited the above text to clarify that these represent the improvements mentioned, and have also added the following sentence to highlight a further improvement.

"Further, the software used for historical anthropogenic emissions and future scenario emissions have been made publicly available as open-source software."

Specific comments:

1. Figure 1. What is the reason for using multiple boxes to represent "Spatially distributed emissions for each country"?

This is, as described in the caption "Emissions data at the level of country and aggregate sector are mapped to spatial grids separately by country and sector". The main text has also been edited as follows to clarify:

"After assigning emissions to spatial grids by country, the spatially distributed emissions from each country, are then added into a global spatial matrix."

2. Figure 2. The legend is too small to identify.

We have increased the text size for the title, labels, and legend. We chose not to increase the label font size too much (these are identical in all figures) so as to leave room for the figure itself.

Editorial comments:
(i) Missing software name and version identifier. Software is provided for generating the data sets. This software needs to be named and versioned, and the name and version included in the paper title. This is important as the methods and software presented may change in the future and it is important that others are able to clearly identify the version that this manuscript describes.

We agree that we should note the specific versions of the software used. We have addressed this as part of the versioning and production of software DOI's as per the comment below.

Please note that the paper describes a methodology for producing these gridded data that uses multiple software systems. We do not, therefore, feel that it would be useful to add the names of all the software systems to the paper title. We will make sure the software is appropriately addressed, including version numbers and (where possible) DOI's.

(ii) Github URLs. Github is an excellent development platform, but it lacks the features required of an archive. GitHub themselves tell authors to use Zenodo for this purpose. The authors should follow the procedure detailed here to archive the exact version of the software used to create the results presented: https://guides.github.com/activities/citable-code/ . The resulting Zenodo repositories present the correct bibliography entries to use.

This is an excellent suggestion, and we have implemented this with DOIs and full references.

(iii) A website is cited for historical burning emissions, this is insufficiently persistent. Instead (or as well), the corresponding paper should be cited. I believe it is: https://doi.org/10.5194/gmd-10-3329-2017.

We have added the appropriate paper reference. We do feel it is useful to point readers to the web site, but we have also cited the appropriate journal paper for this work (e.g., Rabin et al. 2017).

(iv) The reference to the CMIP6 data from the code and data section is too generic to be at all useful. This is only a technical issue, though, because it appears that the exact identifiers are in table 4.

Table 4 should be referenced from the code and data availability section (this may seem petty, but readers of GMD papers know to go to this section to find the data).

Good suggestion. This has been done.

(v) In table 4 itself filenames are given to identify data on ESGF. This is unsafe. The ESGF records themselves provide the correct DOIs to cite, and if the authors click through those DOIs they get the full citation and even corresponding BibTeX and RIS files. The filenames should be replaced with proper data citations.

We have added the full data citations in the data availability appendix, with the full data citation with DOI's in the references, and also noted this in the Table 4 caption. We also provide the filenames in table four, as this is how most users will find the data for the immediate future, and this is more descriptive than a reference or DOI.